# Using Progressive Muscle Relaxation to Increase Academic Engagement for Elementary School Students with Autism Spectrum Disorder

**DOI:** 10.3390/bs15111516

**Published:** 2025-11-08

**Authors:** Lillian McCook, Marissa L. Del Vecchio, Kimberly Crosland

**Affiliations:** Department of Child and Family Studies, College of Behavioral and Community Sciences, University of Sout Florida, Tampa, FL 33620, USAmdelvecchio@usf.edu (M.L.D.V.)

**Keywords:** antecedent intervention, progressive muscle relaxation, academic engagement, autism spectrum disorder

## Abstract

Students diagnosed with autism spectrum disorder (ASD) commonly struggle with self-regulation skills, which can lead to less social inclusion, difficulties with peer and teacher interactions, low academic performance and high levels of challenging behaviors. Alternatively, when students are equipped with strong self-regulatory capabilities, their social development and academic performance is enhanced, leading to improved well-being, increased attention in the classroom, and lower levels of challenging behaviors. Research suggests that the use of progressive muscle relaxation (PMR) may show promising results in improving observable behaviors such as academic engagement and challenging behaviors in the classroom. However, much of the current literature focuses on the positive effects of PMR solely when targeting private events, such as anxiety, executive functioning, and autonomic arousal. The purpose of this study was to conduct a preliminary examination to explore the effects of brief video-based PMR to increase academic engagement in the classroom with three elementary school students diagnosed with ASD. Using visual and statistical analyses, the results from this study showed that PMR showed promising increases in academic engagement across all three participants. Social validity ratings indicated that the teacher and participants were satisfied with the intervention and rated PMR as a feasible and acceptable behavior-management strategy in the classroom. While social validity outcomes were positive, they were limited as they consisted of short Likert-type scale questions completed by one single teacher and three students. Given the small sample size of this exploratory study, future studies should incorporate additional participants and evaluate the long-term impacts of PMR for improving engagement and academic outcomes for students with ASD.

## 1. Introduction

Self-regulation has been described as an overarching construct that involves one’s ability to manage emotional experiences and expressions (i.e., emotion regulation), executive functioning, and approach and withdrawal behaviors ([16]; [32]). Students diagnosed with autism spectrum disorder (ASD) may have unmet needs that result in difficulties in self-regulation ([12]; [27]; [28]). Unmet needs related to communication, sensory sensitivities, and executive functioning could result in students with ASD having difficulty managing emotional responses, leading to the occurrence of different types of challenging behaviors ([23]; [39]). For example, teachers have reported disruptive behavior in the classroom as one of the most common and difficult to manage forms of problem behavior in the classroom ([51]). [39] ([39]) found that when students with ASD engage in high levels of challenging behaviors, their relationship with their teacher is negatively affected, thereby reducing their levels of social inclusion and peer interactions in the classroom. Decreased levels in students’ social interactions have been correlated with low academic performance ([7]; [38]; [50]) and further contribute to increased levels of challenging behavior in the classroom ([39]).

Research has shown that when students are better able to self-regulate, it is more likely that enhanced peer interactions and improved academic performance will be observed, along with decreases in challenging behavior ([1]; [6]; [10]; [55]). The benefits of improved self-regulation can include increased autonomy, positive school engagement, improved peer relationships, and effective sensory regulation ([16]). Therefore, to enhance both learning and social outcomes for students and remediate common teacher stressors, it is imperative to provide students with strategies to better self-regulate their behavior and create a classroom environment that promotes enhanced learning opportunities and improved engagement ([21]). Some self-regulation strategies that have been helpful for students with ASD include self-monitoring, goal setting, self-reinforcement, and self-instruction ([38]). While these strategies can be effective in improving academic outcomes, they also require consistent effort, tracking, and accuracy on the part of the student. This can result in increased anxiety for the student with ASD and an increased teacher burden to ensure adherence and accuracy ([15]). Hence, strategies that involve preventative measures that decrease stress, improve self-regulation, improve academic engagement, and are simple to implement could be helpful in the classroom for students with ASD.

Relaxation techniques, such as yoga ([44]), mindfulness ([45]) and progressive muscle relaxation (PMR; e.g., [17]; [25]) have been used as antecedent interventions to improve attention, concentration, and self-regulatory processes (e.g., emotion regulation, executive functioning, hyperactivity) for students in the classroom. A review by [22] ([22]) of 47 studies using yoga in school settings indicated that improvements were observed across several areas, including greater self-confidence, attention, communication, and social contributions. They suggest that relaxation techniques such as yoga hold promise for enhancing positive student behaviors related to mental health and academic performance. While fewer studies exist focusing on the use of relaxation techniques for ASD, studies have indicated positive outcomes, including increased self-regulation, enhanced attention, and improved communication ([42]). A randomized control trial of yoga and children with ASD in schools conducted by [44] ([44]) showed significant improvements in teacher-reported social communication and imitation skills while also decreasing behaviors that could interfere with learning, including hyperactivity, noncompliance, and inappropriate speech. Several additional studies specific to children and youth with ASD or attentional problems indicated improvements in mood, empathy, teamwork skills, and attention ([35]; [46]). The results of these studies show promising trends in the positive effects of yoga training and mindfulness-based programs for children with ASD, which could serve as additional or complementary interventions to support academic success.

While yoga and mindfulness-based programs have reported some positive results, some limitations have been associated with these techniques that may call the feasibility of these interventions into question. First, in the study conducted by [49] ([49]), yoga training occurred for 1 h during the school day and certified yoga instructors were required to complete each session. Given the significant time constraints already faced by teachers, the duration of these sessions may not feasibly fit into typical classroom routines and may not be maintainable throughout the school year ([4]; [38]). Additionally, the need for a certified yoga instructor may not be a cost-effective method, thereby weakening the contextual fit of yoga-based training in the classroom. Additionally, despite previous research that demonstrated that yoga and mindfulness training produced meaningful changes in student behavior, a more recent study conducted by [43] ([43]) showed that the social validity ratings by teachers and students were low. Teachers in this study indicated that they did not observe changes in behavior for all students and questioned whether the intervention could be generalized to all students and contexts. One student also reported that the intervention was not effective in helping them improve their behavior in the classroom. Therefore, the identification of effective relaxation strategies that support teacher concerns, as well as the time and resource constraints typically faced by schools, is needed to produce meaningful results for students with ASD.

To address this concern, PMR may be a viable antecedent intervention that was used to increase self-regulation skills, improve executive function, and decrease challenging behaviors exhibited by typically developing children (e.g., [17]), adolescents (e.g., [18]), students with ASD (e.g., [30]), emotional behavior disorders (EBD; [25]), and learning and intellectual disabilities (e.g., [48]). PMR is a relaxation technique intended to improve self-control and reduce behavioral dysregulation that may be associated with heightened stimulation ([26]; [54]), and teaches awareness and control of somatic reactions to distress (e.g., anxiety, anger, pain; [20]). Similarly to yoga and other exercise interventions, PMR may result in decreased stress and improved mood, resulting in positive effects on attention, engagement, and concentration in the classroom ([9]). The PMR process involves the systematic tensing and releasing of major muscle groups in the body, including arms, shoulders, chest, legs, feet, hands, abdomen, neck, and face ([26]); and is often accompanied with imagery when it is implemented for children to augment their understanding of the conceptual components of tensing and relaxing each muscle group ([41]). Several hypotheses related to the mechanisms related to PMR that result in improved outcomes are that tensing and releasing muscles will directly affect arousal, sensory regulation, and attention by calming the nervous system. Arousal is regulated by increasing the parasympathetic nervous system, creating a calm state that is optimal for learning. In addition, PMR can decrease heart rate, blood pressure, and respiratory rate, which also results in a calmer state more conducive to class engagement ([52]). For children with autism, PMR could help to mitigate hypersensitivity by lowering anxiety and overall sensory reactivity, which could reduce the impact of triggers in the environment, such as loud noises, thus increasing attention and engagement in the classroom.

In a school setting, PMR is typically implemented by a teacher or trainer, who instruct students to tighten a specific muscle group for 5–10 s and slowly release the tension in that muscle group and repeat the process one to two times ([26]). Typical PMR sessions range from approximately 20 min (e.g., [29]) to 40 min (e.g., [25]). As PMR can be implemented by the teacher, the need for outside individuals (e.g., certified yoga instructor; [49]) and significant time allocation for the implementation of the intervention is reduced, which may help to increase the contextual fit of this relaxation strategy compared to yoga or SoF. [17] ([17]) conducted a randomized control trial to evaluate the effects of PMR on the attention and executive functioning of kindergarten students between the ages of 5 and 6 years old. The results of this study demonstrated that participants in the PMR group showed enhanced attention and improved memory and motor inhibition skills compared to those in the control group; aligning with previous research that supports the effectiveness of self-regulation training to improve academic performance. [24] ([24]) used PMR to reduce test anxiety for 160 elementary-aged students. They found that short-term use of PMR had a significant calming effect, suggesting that children can learn these types of techniques within a relatively brief timeframe. Similarly, [18] ([18]) conducted a quasi-experimental design to evaluate the effects of PMR to decrease exam-induced anxiety, stress, and depression through web-based training sessions with high school students. Statistical analyses revealed significant effects for the experimental group, in which their reported levels of stress, anxiety, and depression decreased following the implementation of the PMR intervention compared to the control group. The results from these studies show that through training, children and adolescents were able to utilize relaxation techniques to improve self-regulation during anxiety and stress-inducing situations.

In addition to studies indicating improvements in anxiety and attention, other studies have shown decreases in challenging behaviors ([30]; [25]; [48]). For example, [30] ([30]) implemented a PMR intervention to decrease the duration of disruptive behaviors exhibited by a 12-year-old male diagnosed with ASD in the home setting. Results showed that when PMR occurred prior to a leisure activity, the duration of disruptive behaviors decreased compared to the duration of disruptive behaviors during leisure activities not preceded by PMR and during baseline. The researchers also evaluated the level to which the participant acquired the skills and independently engaged in PMR following training. Results were positive as they found that the participant acquired most of the steps by the last seven sessions of PMR training and performed an average of 81% of the steps independently (i.e., no prompting) following training. The authors noted that it is likely these skills generalized to novel situations, people and settings, as anecdotal reports indicated that the participant engaged in the PMR steps outside of observational sessions; further supporting the utility and maintenance of self-regulation skill training for individuals with ASD.

In line with [38] ([38]), the use of PMR as a self-regulation intervention has been associated with high social validity ratings from both teachers and students across multiple studies. For example, students reported an increased ability to relax, improved concentration, willingness to perform the exercises at home, and expressed overall enjoyment of the intervention ([48]). Teachers involved in this study also reported an observable increase in students’ prosocial behavior and overall motivation following PMR ([48]). [33] ([33]) used a vignette format to measure teacher acceptance on the use of PMR in the classroom. Participants reported high levels of acceptability for PMR as a school-based intervention, with no statistically significant variations between educational level or years of experience. It is important to note that the sample consisted of teachers from 49 states, suggesting that PMR may be a widely applicable intervention across a variety of environmental contexts ([33]).

Unlike other commonly used relaxation strategies, (e.g., yoga or mindfulness-based curriculums), effective implementation of PMR requires minimal time and training, making it a practical and contextually fit option for classroom management ([26]). [11] ([11]) also suggested that an advantage of PMR as a school-based approach is its flexibility to be taught to individual students with individualized education plans (IEP), or as a collaborative process across larger groups (i.e., class wide). Additionally, the procedures associated with PMR can be easily incorporated within the existing classroom environment, as the length of the procedure can be adapted to accommodate the scheduling needs and typical routines within the school setting ([41]). Despite numerous studies that have demonstrated both the feasibility and effectiveness of PMR, the application of such procedures may still seem overwhelming to many educators, who already experience stress and burnout due to ongoing demands and limited resources ([4]; [38]). Research has demonstrated the effectiveness of technology in improving both student and teacher outcomes in the classroom (e.g., [5]; [14]) and lessening teacher workload ([31]). Video-based PMR lessons have been identified as being just as effective as live PMR instruction ([29]). [29] ([29]) compared the use of live-instruction PMR and immersive video models of PMR to reduce autonomic arousal (e.g., increased heartrate, perseveration, overstimulation). No significant difference between the utility of the two modal variations in intervention delivery was identified, as both produced meaningful results. Interestingly, [29] ([29]) found higher social validity ratings toward video-based PMR sessions compared to live instruction ([29]). Therefore, to further enhance the contextual fit and social validity of PMR in the classroom, the use of a technologically based intervention approach may be a useful tool for both researchers and teachers ([37]).

Careful consideration is warranted related to PMR research as the majority of studies that employed PMR assessed its effects on self-reported events such as anxiety and stress (e.g., [18]; [24]; [48]), while fewer studies have evaluated the effects on observable behavior, such as academic engagement. Studies using self-report measures and rating scales can be biased for a variety of reasons, including social desirability, response bias, and issues with recalling and reporting accurately ([40]). Only one known study collected direct observation data on disruptive behaviors at home, with one participant with ASD ([30]). Additional studies that have more participants and directly observe school-related behaviors such as academic engagement are warranted. Academic engagement has been correlated with improved academic outcomes, including higher achievement and increases in graduation and enrollment in higher education ([57]). Studies have defined academic engagement in different ways, as it can incorporate a variety of learning-related activities in and outside of the classroom (attending class, active participation, completing assignments, enthusiasm with the class content, completing homework; [3]). Studies indicate that directly observing “on-task” behavior such as completing assigned work, answering questions, and listening to instructions may be an effective way to objectively measure student learning, and high “on-task” behavior has been correlated with improved academic outcomes ([8]; [47]). It is important to note that engagement may not be inherently meaningful unless it is aligned with student goals. As studies using PMR have reported improvements in anxiety and stress, it would also be important to determine if PMR results in increases in student academic engagement. One feasible and efficient way to measure academic engagement in the classroom would be using momentary time sampling (MTS), in which observers record the occurrence of engagement if it is happening at the exact end of a predetermined interval ([36]). This method allows for less constant observation and decreased bias in observation, and is ideal for dynamic settings such as classrooms.

As described earlier, only one study was found that evaluated the effects of PMR for one student with ASD ([30]), which used live PMR instruction in the home setting. Therefore, studies using PMR and those that directly measure the academic engagement behaviors of autistic students in the classroom are needed. Furthermore, most research has shown that PMR sessions ranged from 20 to 40 min (e.g., [25]; [29]); however, the implementation of shorter PMR sessions may help to address the time constraints often faced by teachers in the school system. Research is needed to evaluate if a brief video-based PMR intervention would be effective for students with ASD and be feasible for teachers to implement in the classroom. Therefore, the purpose of this study was to conduct a preliminary examination of the use of brief video-based PMR for increasing academic engagement with elementary school students diagnosed with ASD in a classroom setting. A secondary purpose was to determine if teachers and students would report the PMR intervention to be feasible, acceptable, and effective.

## 2. Method

### 2.1. Participants and Setting

Following approval from both university and school district institutional review boards (IRB), consent was obtained from the teacher and each student’s parent or legal guardian. Additionally, assent was obtained from each student participant. During the assent process, the researcher explained the contents of the form, outlined the expectations of the study and encouraged each student to ask questions or express any concerns they had about the study.

This study took place in a self-contained special education classroom within a public elementary school in a southeastern state in the United States. Recruitment flyers were dispersed across the public elementary school that provided information about the study and encouraged interested teachers to contact the first author of this study. Thus, the teacher of the participating classroom contacted the first author to express interest in the study and set up an initial meeting and subsequent observations. During the initial meeting, the first author described the operational definition of academic engagement (see Section 2.4) and asked the teacher to identify, based on her perspective, the academic period in which academic behavior was the lowest (i.e., highest levels of challenging and disruptive behavior exhibited by the students). Students in this classroom ranged from second to fifth grade and were between the ages of 8 and 11 years old. The classroom contained ten students, one teacher, and two instructional aids. In general, students within this classroom were diagnosed with ASD in addition to another developmental and/or intellectual disability. Specific diagnoses for each student within the classroom were unknown, as informed consent was only obtained for the three students who participated in this study. The lack of demographic data, including socio-economic status, and the fact that participants were teacher-selected is a selection bias limitation in this study that should be noted. Observation sessions occurred during the period in which academic engagement was lowest, according to anecdotal reports from the classroom teacher that were obtained in the initial meeting described above. The chosen academic period was English Language Arts (ELA), which occurred at the end of the school day.

The intervention was implemented class-wide; however, only three students were recruited to be directly observed in this study. These students were chosen based on teacher referrals during the initial meeting with the researcher, and observations were collected to ensure they met the inclusion criteria. The teacher was asked to identify students who demonstrated low academic engagement and who might benefit from the PMR intervention. To be included in this study, students had to be enrolled within the school district, attend class on a regular basis (i.e., did not have recurring absences), have a primary diagnosis of ASD, and exhibit low academic engagement based on the teacher’s perspective. Students were excluded from participation in this study if they engaged in dangerous or severe problem behavior (e.g., frequent property destruction, physical aggression towards staff or peers, repetitive self-injurious behavior) or exhibited patterns of recurrent absences from school (e.g., more than five per month).

Pseudonyms were provided to each student who participated in this study to maintain anonymity. Isaac was a 7-year-old White male in the third grade, diagnosed with ASD. He enjoyed reading, coloring, and socializing with his peers. Although his language was developmentally appropriate and matched that of his typically developing peers, he often engaged in repetitive language and stuttering. Isaac was recommended for this study based on teacher reports of consistent disruptions (e.g., calling out, talking to peers during independent work time, and playing with items on his desk), trouble staying on task, and frequent emotional outbursts. Margot was a 9-year-old Black female in the fourth grade and was diagnosed with ASD. Margot had a limited vocal repertoire, that primarily consisted of one-to-three-word sentences, but had strong receptive language demonstrated by her ability to follow through with requests and respond to questions from peers and her teacher. Margot excelled in reading and writing and enjoyed being a classroom helper. Margot was recommended for this study due to frequent out-of-seat behavior, inattentiveness, and difficulties staying on task. Julia was an 11-year-old Hispanic female in the fifth grade. Julia was diagnosed with ASD and frequently engaged in calling out, arguing, protesting, and inappropriate social interactions (e.g., poking, shouting, or laughing at peers). Julia read at a developmentally appropriate reading level, and enjoyed physical activity, choral responding, and iStation (a computer-based comprehensive reading and math e-learning system).

The teacher in this study was a 32-year-old White female with a degree in early childhood education focused specifically on exceptional student education (ESE) classrooms who held academic endorsements in ASD and reading. She had worked in the same classroom for 2 years but also taught general education during summer school. The teacher was recruited for this study based on her willingness to participate in a brief training and on her ability to incorporate the PMR intervention within her daily instruction delivery. Incentives were not provided to the teacher or students who participated in this study.

### 2.2. Materials

The materials used in this study included a Wi-Fi-enabled screen projector (i.e., SmartBoard^®^; SMART Technologies ULC, 2025), a laptop, and a timer. There were three PMR videos that were used during intervention that were found on YouTube and ranged from 4 to 7 min in length (Video 1 = 6 min, Video 2 = 4 min, and Video 3 = 7 min long). The following criteria were used to select the videos: (1) appropriate duration (i.e., under 10 min), (2) demonstrated the tension of each muscle gIroup at least one time, (3) used easy-to-follow language for children. A data sheet that specifically evaluated the percentage of intervals of academic engagement across all phases was created and used throughout the course of this study. During intervention phases, a second datasheet was used to record the percentage of intervals in which participants engaged in the PMR video. A task analysis (TA) and a treatment integrity (TI) checklist created by the researcher was also used in this study.

### 2.3. Experimental Design

To assess the effectiveness of PMR in increasing the participants’ level of academic engagement, this study utilized an ABAB reversal design. Phases included baseline and intervention, as well as a return to baseline and reintroduction of intervention.

### 2.4. Target Behaviors and Data Collection

The dependent variable in this study was academic engagement, defined as the participants’ active involvement in assigned tasks/activity and/or appropriate interactions with the teacher and/or peers (e.g., participation in discussions or activities, following group instructions, maintaining a quiet demeanor and remaining seated while completing tasks during instructional and/or independent work periods, completing assigned work or remaining oriented towards teacher if teacher was presenting information/providing instructions). When academic engagement was not present, the participant was considered off-task (e.g., drawing on their paper, putting their head down on their desk, playing with a toy or any materials that were not required for the specific instruction, moving out of their seat, talking to a peer or laughing, calling out). Research has shown that the use of MTS with short intervals (e.g., 10 s) provides a more accurate representation of student behavior in the classroom compared to extended intervals (e.g., 5 min; [19]). Additionally, [36] ([36]) recommended that MTS measures be used as a behavioral observation method for engagement and challenging behaviors in the classroom, as MTS produces a more representative data set compared to other time sampling methods (e.g., partial interval recording). Therefore, academic engagement was measured using momentary time sampling (MTS) with 10 s intervals during a 20 min observation period. MTS was used in accordance with previous research and to ensure accurate data collection from more than one participant at a time. Video engagement was measured upon the start of the PMR video to evaluate the level to which students attended to and followed along with the video model using MTS with 10 s intervals. A separate datasheet was used for the measurement of academic engagement following the implementation of the PMR video. Video engagement data were used as a measure solely to determine whether or not the observation session would occur for that day. As long as participants’ level of video engagement was above 50% of intervals, observation sessions ensued. If video engagement was below 50% of intervals, observation sessions would not have been conducted; however, this was never the case for any participant across intervention phases. Participants were considered to be engaged with the video when they actively participated in the instructed movements (e.g., were observed squeezing their hands when the video directed them to do so). For both academic and video engagement, at the end of each interval, the researcher scanned the room from left to right and marked the presence or absence of engagement at that moment for each participant in the same sequence. The order of observation for each student remained the same throughout each session to ensure consistency of measurement.

Observation periods for academic engagement occurred immediately following the PMR video for 20 min across all sessions and phases within this study, and occurred in the afternoon during ELA within the same classroom for all participants. If the participant was academically engaged at the end of each 10 s interval, the researcher scored the interval as “+” using the data sheet provided. Conversely, if the participant did not demonstrate academic engagement at the end of the interval, the observer recorded “−” on the data sheet. After each session, the sum of all “+” intervals was divided by the total number of intervals and multiplied by 100 to obtain a percentage of intervals of academic engagement for the session.

### 2.5. Interobserver Agreement

To assess interobserver agreement (IOA), two observers (i.e., the researcher and research assistant) simultaneously and independently recorded the intervals of academic engagement exhibited by each participant throughout the instructional period. IOA was not collected for video engagement. To ensure a shared understanding of the specific topography of academic engagement, the researcher provided the secondary observer with a vocal description of the dependent variable, as described above. The definition of academic engagement was always at the top of each data sheet to limit the likelihood of observer drift. Both observers were required to obtain no less than 90% agreement for the target behavior across three consecutive training sessions. These sessions utilized YouTube videos of students who engaged in similar topographies of the target behavior (i.e., academic engagement). IOA was calculated for an average of 37% of sessions across baseline and intervention phases for each participant. More specifically, IOA was calculated for an average of 40% of sessions for both the first baseline and intervention phase, 39% for the second baseline phase, and 28% for the second intervention phase. Agreement occurred when both observers recorded either a “+” or a “−” in an interval, and a disagreement was observed if one observer marked a “+” and the second observer recorded a “−” in the same interval. An exact agreement measure was used to calculate IOA by dividing the number of intervals where both parties agreed by the total number of intervals and multiplying this number by 100. The average IOA score was 98% across all participants and across all phases (see Table 1 for individual IOA scores).

### 2.6. Treatment Integrity

To ensure fidelity of the intervention, teacher TI was assessed for an average of 80% (i.e., 4 out of 5 sessions) in the initial intervention phase and 50% (i.e., 2 out of 4 sessions) in the second intervention phase. The researcher evaluated TI utilizing a checklist that outlined the six specific steps included in the TA (i.e., inform students that they were going to participate in PMR, provide clear expectations to remain seated, refrain from talking, and follow along with the video, provide students with the opportunity to ask questions, begin the PMR video, sit in front of the classroom while modeling each step demonstrated within the video, thank students for their participation in the video, and begin the next activity). Each step was marked as “Y” if correctly executed by the teacher, and as “N” if errors were observed. The overall TI score was calculated by dividing the number of “Y” responses by the total number of steps and multiplying by 100. TI was 100% across both intervention phases and all participants. Specific session-level data on treatment integrity could not be retrieved and thus is a limitation in this study.

### 2.7. Social Validity

A social validity questionnaire was administered to all participants and the teacher following the completion of this study. A five-item questionnaire was provided to teachers to assess acceptability, feasibility, and overall satisfaction of implementing PMR video sessions in their classroom. The 5-point Likert scale scoring system ranged from 1, indicating *strong disagreement,* to 5, indicating *strong agreement,* with the corresponding statement. The questionnaire also included three open-ended questions for the teacher to provide more information about her experience implementing PMR. Students were also provided with a social validity questionnaire which measured the students’ perceptions of the likeability of PMR, their perceived behavioral changes after participation in the videos, and their interest in continuing to incorporate PMR in the classroom. These data were collected using a three question, 3-point Likert scale where a score of 1 corresponded to a sad face, a score of 2 a neutral face, and a score of 3 a happy face. The students also were given three open-ended questions to answer regarding their experience participating in the PMR video sessions.

### 2.8. Procedures

Data were collected in the afternoon during ELA, which consisted of group reading time, reading comprehension work tasks (e.g., independent worksheets), and/or a class-wide discussion of the corresponding reading. Due to variability in the students’ schedule (e.g., counseling sessions, specials, recess) in the morning and reports of academic engagement being lowest in the afternoons, the time and academic subject in which observation sessions occurred remained consistent throughout this study. Approximately two to four observation sessions occurred per week. During some weeks, both baseline and intervention sessions occurred, while in other weeks, only baseline or intervention sessions occurred.

#### 2.8.1. Baseline

Following identification of the observation time and subject, baseline data collection began. Baseline sessions were conducted during ELA and were 20 min in length. No modifications to the classroom schedule or environment were made during this time, and the teacher was told to carry out the class and respond to student behavior as normal. Baseline data were collected until data were relatively consistent, with minimal variability, or showed a downward trend.

#### 2.8.2. PMR Teacher Training

Following baseline, the researcher conducted a brief training session with the teacher and described the intervention expectations. One training session was conducted during the teacher’s planning period, which lasted approximately 20 min in length. The researcher provided a brief explanation of PMR (i.e., components and behaviors associated with PMR and previous, relevant research). The teacher was offered the opportunity to ask any questions or request clarification related to PMR to ensure she had a comprehensive understanding of the basic components of the behavioral mechanisms of PMR. Next, the researcher succinctly explained and modeled the intervention using the treatment integrity TA, which outlined the specific steps to be completed by the teacher during implementation. The teacher rehearsed each step and was provided feedback as necessary. During this time, TI was assessed to evaluate mastery of each step within the TA. Once the teacher correctly demonstrated each step with 100% fidelity, training concluded.

#### 2.8.3. PMR Intervention

Following the completion of the initial baseline phase and teacher training, the first intervention phase was initiated. Intervention sessions occurred within 1 week of the last baseline session for all participants. Prior to the start of the ELA period, the teacher was instructed to notify the class that a PMR video was going to be played and to provide clear expectations to the students (e.g., remaining seated, actively participating, and refraining from talking), as outlined by the TA provided by the researcher.

Once students were seated and expectations were made clear, the teacher sat in a chair in the front of the room to model the movements that corresponded to the video. At the start of each session, the researcher marked the video number (i.e., 1, 2, or 3) at the top of each data sheet to ensure video rotation and reduce the risk of student habituation. Videos were randomly chosen by drawing a number prior to each session, with a rule that no one video would be viewed more than two times in a row. If a video was picked two times in a row, the number was removed for the subsequent drawing. Video 1 consisted of an animated female character following along with a voiceover that sounded like a child describing how to engage in the PMR activities for each body part while describing how to imagine performing the activities that coincided with the tensing and releasing of different body parts (e.g., pretending to look at something on a high shelf for stretching and relaxing neck muscles). Video 2 was similar to Video 1, as it showed an animated female character demonstrating relaxing and tensing each body part; however, it did not include imaginary examples and instead included small, close-up pictures of the body parts (e.g., a close up of a foot being tensed and relaxed). Video 3 had a real female person (not an animated person) actively engaging in the stretching and relaxing exercises, and she also demonstrated the PMR strategies using props (e.g., a lemon for hand-squeezing, a stuffed turtle for relaxing and tensing the neck). This video also included a countdown in seconds when performing the tensing and relaxing (i.e., 3, 2, 1, done). During all PMR sessions, the researcher was responsible for starting and ending the designated video for each session, per teacher request. The teacher was told to continue to respond to problematic student behavior as she typically would during this time and to not prompt engagement. Participants’ video engagement was recorded using 10 s MTS. If a participant failed to maintain engagement with the video for at least 50% of intervals, data would not be collected for that participant during the observation after the PMR video. However, all participants remained engaged for an average of 70% of the time across both intervention phases, and video engagement was never below 50% for any participant in any session across both intervention phases.

The 20 min observation session began immediately upon the end of the PMR video. The researcher sat in a chair located at the back of the room where all participants could be clearly observed, and academic engagement was evaluated using MTS for 10 s intervals. The teacher was instructed to deliver academic instruction and respond to students as she typically would during the observation session (i.e., no changes to instruction were provided).

## 3. Results

### 3.1. PMR Outcomes

The data obtained from this study indicated that video-based PMR was effective in increasing the level of academic engagement for all three participants across both intervention phases compared to baseline levels.

#### 3.1.1. Isaac

In the first baseline phase, Isaac’s level of academic engagement remained moderately low, with high levels of variability (*M* = 35%; range = 28–43%) across five sessions. Following the initial implementation of the PMR intervention, there was an immediate increase in Isaac’s level of academic engagement compared to the baseline (*M* = 60%; range = 54–72%). Moderate variability with an increasing trend in the initial intervention phase was observed. In the first three intervention sessions, Isaac’s level of video engagement was similar to that of academic engagement. However, an increasing trend was observed with video engagement, despite the decreased levels of academic engagement in the last two sessions of intervention phase 1. Isaac was engaged with the PMR videos for an average of 69% of intervals (range = 56–81%). Upon removal of the intervention and a return to baseline, Isaac’s level of academic engagement decreased (*M* = 47%; range = 44–51%) compared to the first intervention phase across four sessions. Isaac’s percentage of academic engagement in the second baseline phase decreased from intervention levels but remained above initial baseline levels. His level of academic engagement increased by a mean of 12% following the removal of the intervention compared to the initial baseline phase. Moderate variability and a decreasing trend were observed in the second baseline phase. Isaac’s level of academic engagement increased following reintroduction of the intervention, with a mean of 53% of intervals (range = 51–56%). These data remained stable with a slight increasing trend. An initial high level of video engagement was observed upon the reintroduction of the PMR videos, with a decreasing trend across intervention sessions (*M* = 83%; range = 68–92%); however, Isaac’s average level of video engagement in the second intervention phase was higher to that of the initial intervention phase (see Figure 1). Isaac was present for all baseline and intervention sessions across all phases.

#### 3.1.2. Margot

Margot engaged in low levels of academic engagement in the initial baseline phase, with a mean of 38% (range = 26–49%). A clear decreasing trend was observed, with a minor increase in session five. During the first intervention phase, her level of academic engagement immediately increased (*M* = 62%; range = 49–69%) and remained on an increasing trend. A similar trend was observed for Margot’s levels of video engagement, as she engaged with the PMR videos for an average of 75% of intervals (range = 62–84%). Margot’s level of video engagement was always higher than the level of academic engagement during the first phase of the intervention. Upon removal of the intervention, Margot’s percentage of academic engagement decreased (*M* = 47%; range = 36% to 54%); however, a high degree of variability in this phase was observed. Although similar levels of responding occurred upon the reintroduction of the PMR intervention, Margot’s level of academic engagement improved (*M* = 61%; range = 51% to 74%), as demonstrated by an increasing trend across sessions. Margot engaged with the PMR videos during the reintroduction of the intervention at a similar level of response as depicted in the first intervention phase (*M* = 77%; range = 65–95%). Levels of video engagement were higher than that of academic engagement during sessions 14–16, followed by similar levels of response in session 17 (see Figure 2). Margot was absent for one session in the second baseline phase; therefore, only three observation sessions were conducted in this phase.

#### 3.1.3. Julia

Despite the high level of academic engagement observed for Julia in session one, she displayed low levels of academic engagement in the rest of the initial baseline phase (*M* = 36%; range = 25% to 68%). Following the substantial decrease in the percentage of academic engagement from session 1 to session 2, data remained stable across the remaining sessions in the first baseline phase. Following the introduction of PMR, a significant and immediate increase in Julia’s percentage of academic engagement (*M* = 56%; range = 49% to 68%), with moderate variability, was observed. Similar levels and trends were observed for Julia’s level of video engagement during the first intervention phase (*M* = 58%; range = 52–65%). Upon removal of the intervention, Julia’s level of percentage of academic engagement decreased to similar levels (*M* = 39%; range = 34–46%) as those observed in the initial baseline phase. Julia’s level of academic engagement in the second intervention remained comparable to that of the initial intervention phase (*M* = 47%; range = 44–52%) and increased from baseline levels. However, an increasing trend was observed in this phase, as well as an increase when compared to the first baseline phase. Julia’s level of video engagement depicted a similar trend to that of academic engagement during the second intervention phase; however, video engagement was always higher than academic engagement in this phase (*M* = 59%; range = 57–60%; see Figure 3). Julia was absent for one session in both the second baseline and intervention phases; therefore, only three observation sessions were conducted in each phase.

### 3.2. Effect Sizes

Tau-*U* effect size estimates for the percentage of intervals of academic engagement are represented within Table 2. Tau-*U* effect sizes are reported for each participant across all baseline and intervention phases, in addition to the overall effects of PMR on academic engagement for all participants combined. Tau-*U* effect size estimates between 0 and 0.31 represent small effects, 0.32 and 0.84 represent moderate effects, and 0.85 and 1 represent large effects ([34]). Large effect sizes were observed for both Isaac (τ = 0.97) and Margot (τ = 0.88), indicating that there were minimal overlapping data points across baseline and intervention phases. Isaac had one overlapping data point from the second baseline phase to the second PMR phase and Margot had one data point in both the first intervention and second phases that overlapped with two data points in the second baseline phase. A moderate effect size was observed for Julia (τ = 0.61), indicating that there were some overlapping data across all phases, but a positive effect was still observed. Julia had one data point in the second baseline phase that was higher than two data points in the second PMR phase. Across all phases for all participants, the Tau-*U* effect size was 0.83, indicating a large effect size overall. Caution is advised with the overall Tau-*U* effect size as the final engagement percentages ranged from 51% to 74%. While these percentages were higher than the baseline, they still remained relatively low in the intervention phases.

### 3.3. Social Validity Outcomes

Average social validity ratings for all students and their teacher were positive. The mean social validity rating for the teacher was 4.7 out of 5 (see Table 3). She indicated that the intervention was easy to implement, observed an increase in academic engagement, would implement PMR in the future, and would recommend the intervention to other teachers. Across all students, the mean social validity rating was 2.9 out of 3 (see Table 4). The students reported that they enjoyed the videos, felt more focused and relaxed after the intervention, and expressed a desire to participate in the intervention again. Participants indicated that they enjoyed squeezing their facial muscles and engaging in the deep breathing exercises portrayed in the PMR videos.

## 4. Discussion

The purpose of this study was to assess the effectiveness of antecedent video-based PMR at increasing academic engagement with students diagnosed with ASD in an elementary school classroom. For all participants, low to moderate levels of academic engagement were observed in the initial baseline phase, followed by an immediate increase in academic engagement upon implementation of the PMR intervention. The percentage of academic engagement decreased contingent on the removal of the intervention in the second baseline phase, and upon reimplementation of the intervention, an increase in academic engagement was observed. Following both visual and statistical analyses, the results demonstrate the preliminary effectiveness of PMR in increasing academic engagement in the classroom. For both Isaac and Margot, it should be noted that the second baseline averages did not return to similar levels as the initial baseline, indicating that intervention sensitivity might have occurred. The Tau-U large effect sizes should be interpreted with caution given the few data points in each condition and the relatively small overall increases in engagement for some of the intervention sessions for each participant, as these results may not correlate with socially significant changes in student engagement behavior. However, high social validity ratings were obtained from both the teacher and students, indicating that PMR might be considered a useful strategy to be implemented within school-based settings with students with ASD.

This study extends the literature related to self-regulation strategies to improve academic outcomes for students with ASD. Prior studies have mainly focused on self-monitoring types of strategies that included self-reinforcement and goal-setting for improving self-regulation ([38]). PMR may be easier to implement compared to self-monitoring interventions for both the teacher and student in the classroom and result in increases in engagement. This study extended previous research and contributed to the contextual fit of the intervention through the use of shorter PMR sessions. Despite the effectiveness of 40 min (e.g., [25]) and 30 min (e.g., [48]) PMR sessions, the duration of PMR sessions used in the present study averaged 5.5 min and produced similar effects. Thus, the shorter duration of PMR sessions required significantly less time to implement and allowed for minimal time to be taken away from academic instruction in the classroom. Additionally, this study utilized pre-existing and readily available materials (i.e., the classroom SmartBoard and easily accessible videos), which places fewer demands on teachers’ time and effort, further supporting the contextual fit of this intervention. Increased feasibility of interventions can lead to higher social validity and increased fidelity of the intervention ([2]), enhancing the effectiveness of these strategies in the classroom setting. While not empirically tested yet, shorter videos may be more appealing for students with ASD as they might hold students’ attention and require less cognitive load.

Previous school-based studies have examined the impact of PMR on reducing private events such as anxiety and stress (e.g., [18]; [24]; [48]). However, there is a notable gap in research that evaluates the implementation of class-wide PMR interventions aimed to improve overt behaviors (e.g., academic engagement). The current study systematically measured the effect of PMR on academic engagement and demonstrated positive effects of the intervention across all participants. Students from this study also reported feeling more relaxed and focused after the PMR intervention. Consistent with the findings reported by [48] ([48]), while not measured in this study, it is possible that participants experienced increased levels of relaxation, which may explain the increased levels of academic engagement in intervention compared to baseline levels. Future studies should consider collecting both direct observation and physiological measures to determine if engagement increases and stress levels decrease for students with ASD. Collecting these measures may also contribute to a better understanding of how PMR impacts a variety of self-regulation skills with students with ASD, including managing emotions and behaviors, in addition to class engagement.

Finally, a majority of studies that utilized video-based PMR either included this intervention as part of a treatment package (e.g., multimodal technology-based intervention; [37]) or evaluated its efficacy compared to live-instruction PMR (e.g., [29]). This study aimed to assess the utility of PMR on its own through the use of universally available videos (i.e., those found on YouTube) to further increase accessibility of the intervention. [29] ([29]) reported that video-based PMR received higher social validity ratings compared to live instruction. Consistent with these findings, the teachers and students in this study rated video-based PMR as highly feasible and easy to integrate into the academic period. By providing teachers with brief web-based PMR videos, this study may be easily replicated in the future.

This study led to several observations that are worth noting. First, across all participants, there was an immediate increase in academic engagement following initial exposure to PMR. The average increase in academic engagement across all three participants was 19% following the intervention compared to the initial baseline phase; however, Julia exhibited the most substantial increase from session 10 to session 11 when compared to Isaac and Margot. Additionally, Julia’s level of academic engagement in the initial baseline phase was the lowest of the three participants, excluding the outlier that occurred in session 1. Prior to the start of session 1, Julia’s teacher reported that she was, “having a good day” (i.e., demonstrating high levels of both engagement and on-task behavior) and participated in a highly preferred independent reading activity, as reported by the classroom teacher. Although there is no clear explanation for the high level of academic engagement in the first baseline session, it is possible that this outlier may be due to setting events (e.g., increased sleep, positive interactions with teachers or peers) that were outside of the researcher’s control.

Second, while all three PMR videos produced increased levels of academic engagement, video 3 may have been the most effective. Following the implementation of video 3 in session 8 and the last intervention session (i.e., session 18 for Isaac, 17 for Margot, and 16 for Julia), the highest levels of academic engagement across all participants were observed compared to other sessions in the intervention when videos 1 and 2 were played. Several characteristics of video 3 may be responsible for this difference. For example, video 3 had the longest duration (7 min) compared to videos 1 (6 min) and 2 (4 min) and featured an adult female using props (e.g., squeezing lemons and stuffed animals) as a way to model the tension and relaxation of each muscle as a PMR strategy. Videos 1 and 2, however, consisted of animated characters that vocally described the steps in the PMR (i.e., the specific body part to tense and relax), without the inclusion of other materials that further demonstrated how to accurately tense and relax these muscle groups. Thus, it is possible that video length may have influenced students’ academic engagement, given that they had the opportunity to engage in more PMR activities compared to those of a shorter duration. Additionally, the presence of a person using materials to demonstrate relatable ways to engage in PMR compared to animated characters that simply described the PMR strategies may enhance students’ understanding of how to correctly engage in PMR strategies, leading to increased levels of academic engagement following the video. Moreover, as this was their third exposure to the intervention, this suggests that more practice may be needed to grasp the required steps and to fully benefit from the PMR videos. The sudden decrease in engagement observed across all participants in session 9 may further support this notion, as the video employed in this session (i.e., video 2) was shorter than the video used in session 8 (i.e., video 3). It may be that, for students with ASD, the concrete realistic representation and additional modeling in video 3 aligned with the processing skills of the students when compared to the more abstract animated character versions in videos 1 and 2. Future research should specifically evaluate if certain characteristics of videos are more or less effective for students with ASD.

Lastly, and possibly the most noteworthy observation, was the variability of academic engagement observed across baseline and intervention phases. The mean level of academic engagement across all participants was 36% and 45% in the initial baseline phase and second baseline phase, respectively. Although a decrease in the percentage of academic engagement from intervention occurred in the second baseline phase for all participants, the difference in participants’ levels of academic engagement across baseline phases may be due to variations in ELA activities across each phase. For example, during the initial baseline phase, data were collected primarily during independent reading time. Although the teacher reported that all three typically enjoyed this activity, independent work led to more opportunities for off-task behavior compared to small group activities with the teacher. In the second baseline phase, observation periods occurred during small group activities and worksheets, allowing for more opportunities for participation and academic engagement. These variations were also observed across intervention phases. The final intervention phase produced lower levels of academic engagement overall, compared to the first intervention phase, which may be attributed to schedule changes (e.g., the end of the school year, testing, student absences). For both Isaac and Margot, it should be noted that the second baseline averages did not return to similar levels as the initial baseline. This could be due to the differences in activities described earlier or might be a result of intervention sensitivity. However, despite the differences observed across phases, PMR still resulted in higher levels of academic engagement compared to baseline levels.

Although the present study further contributed to the literature related to the effectiveness of PMR within the classroom, several limitations are worth mentioning. First, it is possible that teacher bias may have influenced the selection of the participants who were recruited and enrolled in this study. Although objective descriptions of participant eligibility criteria and an operational definition of academic engagement were provided, the researchers were unable to control for any selection bias that may have occurred. Next, although the operational definition of academic engagement attempted to include all behaviors associated with a student being engaged with academic instruction, it is possible that our definition may have inadvertently captured behaviors that align more so with those associated with “compliance,” rather than genuine learning. In the future, delineating behaviors associated with genuine learning and engagement (e.g., task completion, task accuracy) and those that are incompatible with genuine learning (e.g., disruptive behaviors), may reduce the risk of overgeneralization of these behaviors. Time constraints (e.g., difficulties with recruitment, end-of-year testing) and frequent classroom schedule changes (e.g., field trips, teacher and student absenteeism, classroom and school-wide events) led to fewer observation sessions across phases. Further, the lack of time also led to an inability to evaluate the level to which these results were maintained in follow-up sessions. The limited number of observation sessions, specifically in the second baseline and intervention phases, may not allow for accurate representation of the effects of the PMR intervention. However, given that there was a minimum of three data points within each phase across all participants, this study still met the guidelines set forth by the What Works Clearinghouse 5.0 standards ([56]), with reservations. The extension of this study’s timeframe would have allowed for increased exposure to the videos, which may have led to higher levels of academic engagement and enhanced the effectiveness of PMR in the classroom.

Another possible limitation that might be present with this study is the brevity of the PMR videos used in this study. Previous research has implemented PMR trainings that are approximately 20–40 min in length (e.g., [25]; [29]; [48]), whereas this study utilized videos between 4 and 7 min in length. It is possible that students did not have the opportunity to learn or engage in the skills for a long enough period of time for the effects of PMR to be completely observed. Future research might incorporate PMR videos more frequently throughout the school day and/or week, rather than once per day, three to four times a week, as conducted in this study and evaluate the long-term use of the PMR intervention. It may be beneficial for future researchers to evaluate the effects of differing durations of PMR training to determine those that lead to the strongest effects. Further, the period of the school year may have led to lower levels of academic engagement during intervention phases, as the school year completed about two weeks following the conclusion of this study. Future research should evaluate the effects of PMR in the classroom to increase academic engagement in the beginning or middle of the school year to ensure that teachers and students are more likely to be engaged in academic material and follow a more consistent schedule. Future research may also evaluate whether the effects of PMR generalize outside of training sessions as a form of self-regulation, and whether the relaxation technique is self-applied, without an external change agent. It may be interesting to measure whether students conduct the PMR steps independently, and if these behaviors lead to increased student engagement in the classroom.

While IOA data were collected for, on average, over 30% of the sessions for each participant (which is a recommended minimum percentage; [13]) and were always more than 96% accurate, it would be beneficial to have a greater number of sessions with a second observer to ensure observer drift does not occur. This can be difficult in school settings, as having additional observers in the classroom can be distracting to both the teacher and students. Both the teacher and students completed social validity questions using Likert-type scales, which could limit the types of important information that was collected related to the acceptability and feasibility of the intervention. Future studies might consider asking additional open-ended questions, obtaining narratives, or conducting structured interviews with teachers and students to obtain richer qualitative data. Additionally, while teacher and student social-validity outcomes were overwhelmingly positive, bias related to teacher investment in the intervention and student motivation to please researchers could result in the over-reporting of positive outcomes. Treatment integrity data remained high for the implementation of the intervention, suggesting that it was feasible and easy to implement in the classroom. Unfortunately, we could not retrieve session-specific information on components that the teacher missed, which is a limitation.

Finally, although the effects of PMR were large when evaluated across all participants and phases, a moderate effect size was observed for Julia, indicating that there was some overlapping data across baseline and intervention phases. In addition, academic engagement did not reach levels close to 100% for any of the three participants, indicating that PMR may be an easy, efficient intervention for teachers to help improve engagement, but additional strategies may be needed to reach high levels of academic engagement. Future research may address this limitation by incorporating reinforcement contingencies (e.g., behavior-specific praise or class-wide reinforcement systems) to promote increased levels of academic engagement, following the implementation of the PMR videos. Additionally, the integration of additional antecedent strategies (e.g., clear expectations, visual schedules) within the classroom routine may foster a more interactive and collaborative approach, which may further enhance the effectiveness of the PMR intervention. Increased exposure to the intervention and the use of additional strategies to promote academic engagement may lead to greater intervention effects and should be further evaluated.

Furthermore, the presence of disruptive behavior (e.g., screaming, elopement, property destruction) exhibited by the participants’ peers may have led to decreased levels of academic engagement across all phases. Peers’ challenging behaviors caused interruptions in the intervention (e.g., delays in starting the videos) and subsequent ELA activities and may have led to lower levels academic engagement among the participants. Future studies may evaluate the efficacy of the PMR videos for students with ASD in an inclusive general education classroom where problem behaviors may be less prevalent. Consistent with the notion posed by [25] ([25]), it is possible that PMR may be effective with other populations (e.g., EBD, ADHD) and should be further evaluated in future research. Given the exploratory nature of these findings, future research should further evaluate the use of brief video-based PMR with a larger and more diverse sample of students with ASD to extend and provide additional insight on the effectiveness of this antecedent school-based approach.

## Figures and Tables

**Figure 1 behavsci-15-01516-f001:**
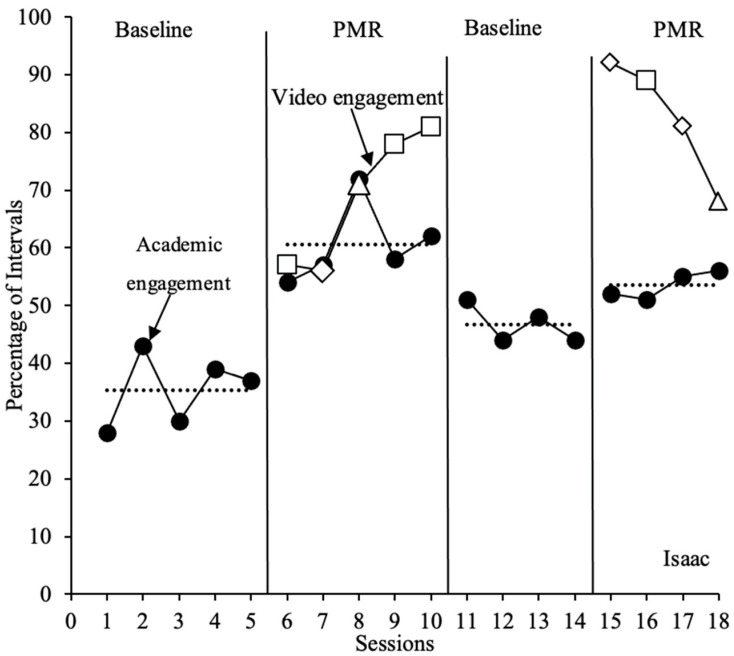
Percentage of academic engagement across 20 min sessions for Isaac. The dotted lines represent the mean percentage of academic engagement across each phase. The closed black circles represent academic engagement. The open data markers represent video engagement. Open squares represent sessions in which PMR Video 1 was implemented, open diamonds represent sessions in which PMR Video 2 was implemented, and open triangles represent sessions in which PMR Video 3 was implemented.

**Figure 2 behavsci-15-01516-f002:**
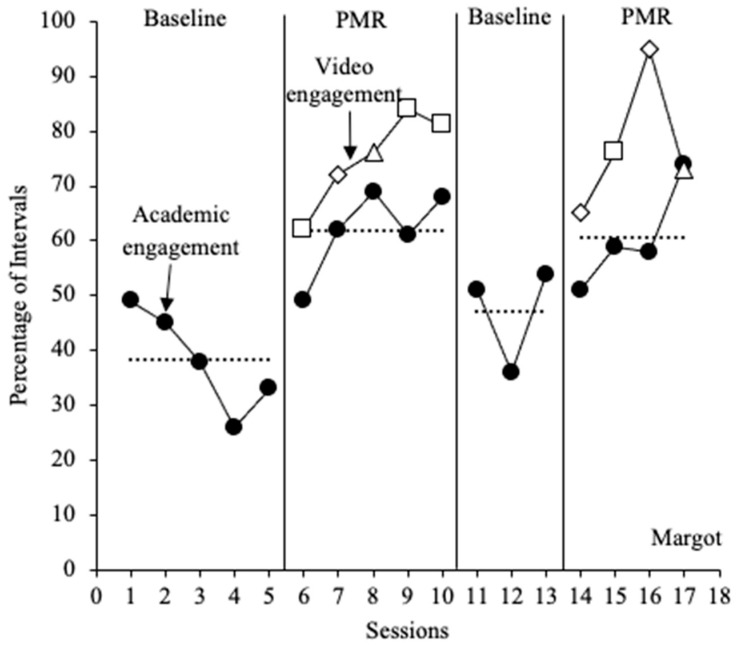
Percentage of academic engagement across 20 min sessions for Margot. The dotted lines represent the mean percentage of academic engagement across each phase. The closed black circles represent academic engagement. The open data markers represent video engagement. Open squares represent sessions in which PMR Video 1 was implemented, open diamonds represent sessions in which PMR Video 2 was implemented, and open triangles represent sessions in which PMR Video 3 was implemented.

**Figure 3 behavsci-15-01516-f003:**
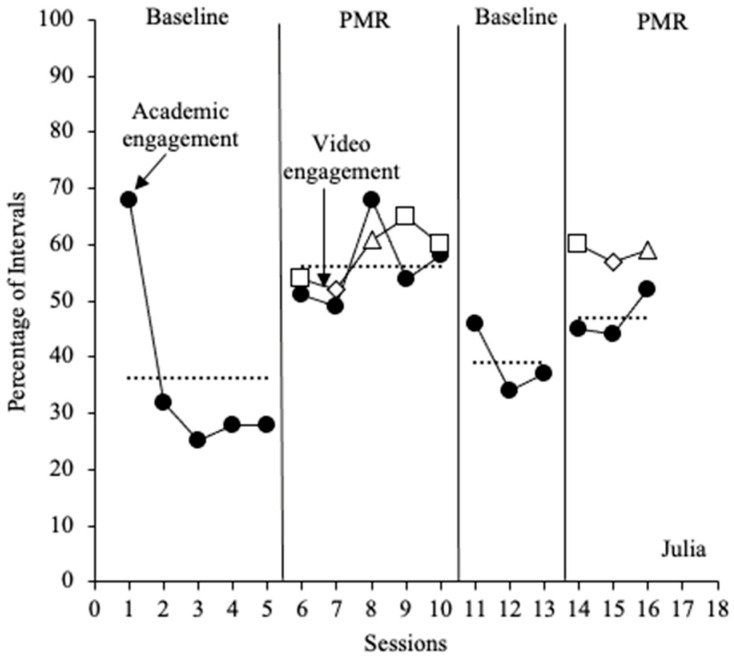
Percentage of academic engagement across 20 min sessions for Julia. The dotted lines represent the mean percentage of academic engagement across each phase. The closed black circles represent academic engagement. The open data markers represent video engagement. Open squares represent sessions in which PMR Video 1 was implemented, open diamonds represent sessions in which PMR Video 2 was implemented, and open triangles represent sessions in which PMR Video 3 was implemented.

**Table 1 behavsci-15-01516-t001:** Interobserver agreement across all participants.

Participant	Phase	IOA Score *M* (Range)	Sessions IOA Assessed (%)
Isaac			
	Baseline 1	98.5% (98–99%)	40%
	Intervention 1	97.0% (96–98%)	40%
	Baseline 2	98.0% (98–98%)	50%
	Intervention 2	99.0% (99–99%)	25%
Margot			
	Baseline 1	98.5% (98–99%)	40%
	Intervention 1	99.0% (98–100%)	40%
	Baseline 2	98.0% (98–98%)	33%
	Intervention 2	98.0% (98–98%)	25%
Julia			
	Baseline 1	97.5% (96–99%)	40%
	Intervention 1	98.5% (98–99%)	40%
	Baseline 2	96.0% (96–96%)	33%
	Intervention 2	98.0% (98–98%)	33%

Note. This table represents the mean IOA scores and number of sessions in which IOA data were collected per phase for each participant.

**Table 2 behavsci-15-01516-t002:** Tau-U effect size estimates for percentage of intervals of academic engagement from baseline and intervention phases for each participant.

Participant	Tau-*U*	90% CI	*p*
Isaac	0.97	[0.64, 1.00]	0.0000
Margot	0.88	[0.53, 1.00]	0.0000
Julia	0.61	[0.25, 1.00]	0.0049
Overall	0.83	[0.63, 1.00]	0.0000

Note. Tau-*U* effect size estimates represent the aggregate amount of non-overlap between both baseline and intervention phases for each participant. The 90% confidence interval (90% CI) represents the range of effect size when replicated. *p*-values (*p*) represent the statistical significance of the effects of the intervention. Values less than 0.01 represent a statistically significant effect, while values of 0.05 or higher indicate that intervention did not produce a statistically significant effect. Calculations within this table were conducted according to [53] ([53]). Baseline was not corrected for any calculation within this table. Moderate to large effects were observed for all participants. Tau-*U* effect size estimates between 0 and 0.31 represent small effects, 0.32 and 0.84 represent moderate effects, and 0.85 and 1 represent large effects ([34]).

**Table 3 behavsci-15-01516-t003:** Teacher social validity ratings.

Teacher Items	Rating
The PMR video sessions were easy to implement in the classroom during the instructional period.	5
After the PMR video sessions, students showed decreases in disruptive behavior.	4
After the PMR video sessions students were more engaged in activities/academics.	4
I would implement PMR in my classroom in the future.	5
I would recommend PMR to other teachers to use in their classroom.	5
Overall, I enjoyed participating in this study.	5
Mean Overall Social Validity Rating (out of 5)	4.67

Note. This table represents teacher social validity questionnaire responses. For teacher items, 1 = Strongly Disagree; 2 = Disagree; 3 = Neutral; 4 = Agree; 5 = Strongly Agree.

**Table 4 behavsci-15-01516-t004:** Student social validity ratings.

Student Items	Rating		
	Isaac	Margot	Julia
I liked the PMR videos.	3	3	2
I could focus better on my work after the videos.	3	3	3
I want to keep using the videos.	3	3	3
Mean Overall Social Validity Rating Per Student	3	3	2.67
Mean Overall Social Validity Rating	2.9		

Note. This table represents student social validity questionnaire responses. For student response items, 1 = sad face; 2 = neutral face; 3 = happy face.

## Data Availability

Data for this study were collected by the first author in partial fulfillment of the requirements for the Master of Science degree in Applied Behavior Analysis at the University of South Florida. Data are available upon request from the corresponding author.

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
