# Peer review of "Using Progressive Muscle Relaxation to Increase Academic Engagement for Elementary School Students with Autism Spectrum Disorder"

_behavsci, 2025, doi:10.3390/bs15111516_

Round 1

Reviewer 1 Report

Comments and Suggestions for Authors

Thank you for the opportunity to review the manuscript, “Using Progressive Muscle Relaxation to Increase Academic Engagement for Elementary Students with Autism Spectrum Disorder”, submitted for consideration of publication in Behavioral Sciences. Though I believe there is some merit to the idea behind this study and appreciate the effort to deliver an intervention that is feasible and acceptable for use within the natural flow of instructional activities for a teacher in a self-contained classroom for students receiving special education services, I have several concerns about the current presentation of this work and its readiness for publication. I’ve offered some specific suggestions below for certain sections. In addition, I’ll offer some broad considerations that contribute to my reservations with this manuscript as is. First, the introduction and discussion seem to try to position this study as one that contributes to the evidence base for use of PMR under a broad array of conditions in schools. Given my questions about the rigor of the methods, I encourage the authors to reposition their work as a preliminary examination of use of PMR within the context created by their specific study rather than adding to a broader evidence base for PMR. Second, questions about the rigor of the methods stem from many assumptions that need to be made based on general descriptions provided by the authors. There are a number of additional details needed to promote replicability and a more thorough evaluation of the authors’ findings. Finally, academic engagement is an important construct with theoretical grounding and operationalized measurement that varies widely in the literature, especially when the focus is on students with varying types of disabilities. As part of reframing this study, I encourage the authors to carefully consider what academic engagement means for students with ASD and the theoretical link they are conceptualizing between PMR and academic engagement for students with ASD. There seems to be some mechanism they believe PMR is tapping into that contributes to their conceptualization of what academic engagement looks like, which should be made more clear. Related to that point, please give careful consideration of the actual operationalization of academic engagement versus not engaged. There are “compliance” focused behaviors (quiet, sitting calmly) that some would argue are less about “academic engagement” and active learning and more about maintaining certain behavioral expectations within academic routines. I believe it is important to be explicit about that when engaging in this type of work and how we expand our understanding of what contributes to improving actual learning and academic performance for students with ASD.

Introduction

Consider broad reorganization to enhance the flow of information that establishes the rationale for this particular study. This is consistent with my recommendations above.

Method

Page 4, line 188: Edit the heading to be “Method” rather than “Methods”, per APA guidelines

Page 4, Participants and Setting section:

  • Please begin this section with confirmation that this study was reviewed and approved by the institutional IRB rather than where it is currently provided on page 5, line 205.
  • When describing the setting in which this study took place, provide additional information about the students as a whole, especially given the intervention was implemented class-wide. Where all students identified with ASD or a broad range of disabilities? Why was this particular classroom selected? How was academic engagement described to the teacher to inform the anecdotal report that led to the selection of English Language Arts being selected as the focus and what did “lowest” academic engagement look like for that classroom? These details are important for replicability and consideration of the inferences made about the intervention.
  • Pg 5, line 198 the authors state that “observations were collected to ensure they met inclusion criteria”, though readers have not been given information about the actual inclusion criteria. The authors go on to describe exclusion criteria, but what was the basis for teachers to “identify students who demonstrated low academic engagement and who might benefit from the PMR intervention”? Providing operationalized descriptors of these criteria is needed to support replication and evaluation of inferences. This information would also provide a better basis for why certain descriptive information is shared about the three student participants.
  • Pg 5 line 231: This is the first and only use of “ESE”. Write out what that means.

Pg 5, Materials section: This section exclusively describes the intervention, so I’d encourage moving this section to the procedures section in which the intervention is described more fully. In the current location, the information provided just raises questions about the delivery of the intervention. The authors also mention use of a task analysis and a treatment integrity checklist in this section, which is definitely a positive, but no other information is provided about those in proximity to those statements. Some reorganization of what information is shared where would help the flow.

Pg 6+ Sections 2.3 – 2.6: I encourage the authors to reorder how information is provided in the method section and move these sections to after section 2.7 describing the overall design and procedures. As a reader, having the contextual information about the procedures associated with the different phases of the study is helpful to understanding and evaluating the data collection methodology.

Pg 8+, Sections 7+: I encourage the authors to again consider their descriptions of their procedures with consideration of replicability throughout this section. For example, stating that “baseline data were collected until data were stable” would benefit from more explicit details about “what data” and how was “stable” determined. Referencing the TA described elsewhere when describing the teacher training could be improved by moving the description of the task analysis and treatment integrity checklist to this section so readers have that information in one place when learning about what the teacher was asked to do as part of implementing the intervention. As another example (pg 9, line 376) references “video rotation”. As varying stimuli may have impact on experimental control (which is later confirmed in the discussion section), please be specific about what that means and how any implications of different stimuli were examined/managed (randomized, counterbalanced, systematically examined for differential responding).

Discussion

Pg 14, line 561+: In discussing the differential impact of video three and differences in the characteristics of that video relative to the to the other two, I encourage the authors to provide further reflection on how the characteristics of their participants may also play a role. For students with ASD, the concrete realistic representation and modeling provided in video three versus an abstract animated representation may intersect with the processing skills of the participants. Given the focus on this study to extend prior research to use of PMR with students with ASD, I encourage the authors enhance their discussion with greater attention to implications specifically for students with ASD rather than a general use of PMR in school settings.

Author Response

Thank you for your careful review and recommendations for our manuscript. We have incorporated edits from all reviewers and have attached a table with each reviewer comment, how it was addressed, and the location to find the edits in the paper. We appreciate your time in reviewing our paper and we think the paper has been substantially strengthened after incorporating the recommendations from you and the other reviewers. 

Reviewer 2 Report

Comments and Suggestions for Authors

Thank you for the opportunity to review the manuscript, “Using Progressive Muscle Relaxation to Increase Academic Engagement for Elementary School Students with Autism Spectrum Disorder.” The authors sought to conduct a progressive muscle relaxation (PMR) procedure prior to a language arts lesson for three participants to evaluate its effects on academic engagement. My recommendation for this paper is to accept after minor revisions. I have outlined my suggestions below:

One strength of this study I found was the use of momentary time sampling. I would have liked to see some references to momentary time sampling, especially in school-based settings in the introduction to set up the author’s selection of this measurement.

I found there to be a lot of detail about the intervention for this study missing. To be technological and to ensure future researchers can replicate this study, I encourage the authors to include more detail about their independent variable. I have outlined a few questions and things I would like to see below:

  1. The authors mention that three different videos were used. They mention in the discussion that these were Youtube videos (universally available). I think this should be introduced earlier in the Method. The authors should also describe the videos (i.e., How did they differ? What skills were targeted, etc.). Also, how were the different videos assigned to each session? Were they randomly assigned or alternated (i.e., Session 1 - video 1, Session 2 – video 2, Session 3 – video 3, Session 4 – video 1 …. ).
  2. How long were videos?
  3. The authors mention the teacher modeled the movements in the video but was that all they did? What happened if a participant did not engage? Was there any prompting? What did this prompting look like, if so.
  4. Did the authors collect engagement data or problem behavior during the videos? It would be interesting to know how engaged they were. Additionally, if the authors have data on problem behavior post PMR video, those data would be interesting as well.

Results

  1. When evaluating the effects of the intervention both visually and the Tau-U data, there is clearly an effect. However, I would like the authors to discuss a bit more about how these data might reflect socially significant changes given they were still relatively small increases and still low overall. For example, in the final phase, Isaac was engaging in an average of 53% engagement, Margot was 61% and Julia was 47%. Do the authors have data on
  2. When looking at the graphs, Isaac had an extra session in the second BL phase and Julia had one less session of final PMR phase. Given this was a group intervention (i.e., all participants received each condition at same time), were these different because of absences or was this a mistake? The authors briefly discuss time constraints in the limitations section but given the differences in session numbers across participants this would be helpful in the results.

Author Response

(The authors gave the same response as above.)

Reviewer 3 Report

Comments and Suggestions for Authors

Thank you for the opportunity to read your manuscript "Using Progressive Muscle Relaxation to Increase Academic Engagement for Elementary School Students with Autism Spectrum Disorder." The manuscript is very well written and clear. As the journal is international, it may be necessary to add the country (assuming the United States) as context (e.g., line 191 "a southeastern state in the United States").

I have a few suggestions to consider in a revision. Do you have the IQ for the participants? As ASD manifests very differently for individuals, it would be helpful to include as much information as possible regarding measured intelligence and communication styles. Why were these three students selected? The inclusion criteria is included but appears to be very broad- off task, not self injurious and attends school regularly. 

Within the baseline and intervention, how long was it between sessions? I am assuming all students were observed on the same days/lessons but it may be helpful to make that clear in the narrative.

The ratings in Table 4 don't match the description in the narrative starting line 477.

While I believe the Discussion may need more attention, the authors make several important points in the Discussion which are important to recognize. For example, the first point is that this study contributes to the literature as it was whole class, but your study only collected data for three students out of the 10 in the class.  Can you say this contributed as a whole class study? Your point about the video differences may warrant a change to how the data within the  graphs are presented to demonstrate which video was used during intervention for each data point.  The number of data points in the second phase limits the strength to which you can claim effectiveness.  Just a thought- is it important to recognize the sensitivity of the intervention as demonstrated during the second baseline? Data from two students didn't return to the initial baseline numbers. While the authors recognize the activities could influence that, recognition in the Discussion section about intervention sensitivity may also be warranted.  I believe the context of the classroom is extremely important and more research within such a classroom is also warranted. It would also be very interesting if the observer could capture the independent application of PMR during instruction outside of the intervention videos. That would strengthen the connection between self regulation (student self application of PMR) and student engagement. As this study just taught the student to perform PMR, future studies should include measurement of student use. 

Again, thank you for the opportunity to read about your work. The clarity of your writing is notable. 

Author Response

(The authors gave the same response as above.)

Reviewer 4 Report

Comments and Suggestions for Authors

1. Brief Summary of the Article

The manuscript evaluates the effects of a brief, video-based progressive muscle relaxation (PMR) intervention on academic engagement in elementary school students diagnosed with autism spectrum disorder (ASD). Conducted in a self-contained special education classroom, the study used an ABAB reversal design with three student participants. Academic engagement was measured through momentary time sampling, and social validity data were collected from both students and the teacher. Results indicated that PMR increased academic engagement with moderate to large effect sizes and received positive social validity ratings, suggesting the feasibility of incorporating PMR in classroom contexts.

The study addresses a relevant issue in special education research, namely how antecedent interventions may promote engagement and self-regulation. The focus on video-based PMR is novel and potentially practical for classroom use. However, there are conceptual, methodological, and interpretive aspects that could be strengthened to increase rigor, generalizability, and alignment with broader debates in autism research.

2. General Comments on Conceptual Framing

The manuscript presents PMR as an effective and broadly feasible strategy to improve academic engagement. While the intervention’s simplicity and acceptability are clear strengths, the conceptual framing could be expanded in several areas:

  • Positioning within self-regulation research: The paper emphasizes the deficits model of self-regulation in ASD but pays less attention to individual variability, alternative explanations for engagement, or strengths-based perspectives. Incorporating literature that considers adaptive regulation strategies used by autistic students would balance the framing.

  • Consideration of neurodiversity perspectives: The discussion frames PMR primarily in terms of reducing “challenging” or “disruptive” behaviors. Including reflection on the potential benefits for well-being, autonomy, or sensory regulation—rather than only compliance or engagement—would align better with current inclusive approaches.

  • Scope of outcomes measured: The study measures only observable engagement and does not examine emotional states, stress reduction, or long-term maintenance. Since PMR is theorized to reduce arousal, measuring physiological or subjective outcomes would provide stronger evidence of mechanism.

Overall, the manuscript presents promising findings but could better situate them within broader theoretical and ethical considerations.

Abstract (Lines 10–29)

  • Lines 16–24: The abstract presents PMR as an “effective strategy” with “moderate to large effect sizes.” This language overstates certainty given the very small sample and lack of long-term data. Suggest softening claims (e.g., “showed promising increases”) and explicitly note the study’s exploratory nature.

  • Lines 25–27: Social validity is mentioned positively. However, the measures were limited (short Likert scales, single teacher). It would be more transparent to acknowledge this limitation in the abstract.

Introduction (Lines 31–187)

  • Lines 32–47: The framing portrays self-regulation primarily as a deficit and links ASD directly with disruptive behaviors. This risks reinforcing deficit-based narratives. Suggest rephrasing to emphasize heterogeneity in self-regulation profiles and to acknowledge that “off-task” or “non-compliant” behaviors may reflect unmet needs rather than inherent deficits.

  • Lines 60–77: While yoga and mindfulness studies are reviewed, the transition to PMR is abrupt. Authors should explicitly contrast PMR with these alternatives, highlighting unique features (e.g., ease of implementation, reduced training demands).

  • Lines 78–112: The literature review summarizes many PMR studies but does not critically appraise their quality. Several were case studies with limited generalizability. Suggest acknowledging methodological limitations to avoid overstating the evidence base.

  • Lines 183–187: The purpose statement is clear but assumes that academic engagement is universally positive. Authors might add nuance, e.g., recognizing that engagement must be meaningful for the learner and not just compliance with tasks.

Methods

Participants and Setting (Lines 189–236):

  • Only three students were included, all teacher-selected. This may bias results toward those most likely to benefit. Authors should acknowledge this limitation explicitly in this section, not only in the discussion.

  • Demographic description focuses on race/ethnicity and language abilities, but socioeconomic context is absent. This omission limits the cultural contextualization of findings.

Materials (Lines 237–248):

  • The intervention videos were sourced from YouTube. This raises concerns about reproducibility: videos may change or disappear, and their quality is not standardized. Authors should either provide stable links in supplementary material or describe video content in sufficient detail for replication.

Target Behaviors and Data Collection (Lines 249–284):

  • The operationalization of “academic engagement” is appropriate but broad (e.g., “remaining seated” is coded as engagement). This risks conflating compliance with genuine learning. Suggest refining the definition or acknowledging this limitation.

  • PMR engagement was used as a gatekeeper (≥50% engagement to proceed with observation). This could bias outcomes, as low-engagement sessions were excluded by design. Authors should address this potential inflation of results.

Interobserver Agreement (Lines 285–305):

  • IOA was collected for ~37% of sessions. Although agreement was high, the relatively small percentage may not sufficiently guard against drift across phases. Suggest discussing this limitation explicitly.

Treatment Integrity (Lines 309–321):

  • TI was reported as 100%, but only sampled in a subset of sessions. A fuller report on which sessions were assessed (e.g., early vs. late in intervention phases) would increase confidence.

Social Validity (Lines 322–336):

  • Teacher data are detailed, but student data are limited to a 3-point Likert scale and minimal open-ended responses. This constrains interpretability. Suggest adding richer qualitative data (e.g., student narratives or classroom observations) in future iterations.

Results (Lines 391–486)

  • Figures 1–3 (Lines 412, 427, 445): Graphs show variability across phases, but text interpretation emphasizes only improvements. For instance, Julia’s data show considerable overlap between baseline and intervention. Authors should present these nuances rather than generalizing positive effects.

  • Lines 447–470: Tau-U analyses are appropriate, but effect size interpretation should be tempered. With only three participants, confidence intervals are wide and may not generalize.

  • Lines 471–486: Social validity outcomes are reported as overwhelmingly positive. Potential demand characteristics (teacher invested in intervention; students motivated to please) should be acknowledged.

Discussion (Lines 487–639)

  • Lines 500–546: Authors emphasize feasibility and brevity of PMR. While valid, claims about broad applicability are premature given the highly specific context (one teacher, one classroom). Suggest reframing to “preliminary evidence of feasibility” rather than generalizing.

  • Lines 547–583: Explanations for variability (e.g., “good day”) are anecdotal and speculative. Authors should either provide stronger evidence for these interpretations or acknowledge uncertainty.

  • Lines 602–639: Limitations section identifies time constraints and lack of follow-up but does not discuss the impact of excluding sessions with <50% video engagement, nor the teacher-selection bias in participant recruitment. These are critical omissions.

Conclusion (Lines 649–669)

  • The conclusion appropriately highlights the promise of PMR, but omits cautionary notes. Authors should briefly emphasize the exploratory nature of findings and the need for replication with larger, more diverse samples.

Author Response

(The authors gave the same response as above.)

Round 2

Reviewer 1 Report

Comments and Suggestions for Authors

Thank you for the opportunity to once again review, “Using Progressive Muscle Relaxation to Increase Academic Engagement for Elementary School Students with Autism Spectrum Disorder”, submitted for consideration of publication in Behavioral Sciences. I was a reviewer on the previous submission. Generally, the authors have been responsive to the prior review, though there remain a couple of opportunities to further strengthen the presentation of their work prior to publication.

Introduction

The authors begin is a definition of what is meant by self-regulation that incorporates multiple types of skills, but the rest of the introduction seems to focus on the outcome of poor self-regulation skills (i.e., challenging behavior). As was suggested in the prior review, I think the opportunity still needs to be addressed to reframe the introduction to focus on strategies to enhance specific aspects of self-regulation and how relaxation techniques factor into what is and isn’t know about each aspect of self-regulation as it relates to academic engagement. As is, the primary rational for this study remains just on addressing challenging behavior.

Pg 2 line 46: Please remove the term “meltdown”. This is a colloquial term that should not be used in published literature.

Method

In my opinion, the authors have been responsive to prior feedback and requests for additional details.

Results

Generally, the results section has been strengthened through the authors’ changes in response to the prior review. However, in describing the social validity data, I am a little confused about the in-text description of the student ratings (stating the ratings were consistently positive) and Table 4 in which the note states that a rating of 3 represents a sad face. The table would suggest the students are consistently negative towards the videos. Please confirm what is correct and adjust accordingly. This was identified in the prior review but not fully addressed.

Discussion

The authors provided a more nuanced discussion of the findings in response to requests in the prior review. What strikes me in this revised version is that there is no attempt in the discussion to bring their findings back to self-regulation. The focus stays exclusively on academic engagement. With some continued reservations about the framing of the introduction, I encourage the authors to consider how self-regulation should be addressed more fully in the discussion section relative to the implications of their study and prior work, or, refocus the introduction just on active engagement by reflecting on observable behavior and internal processes that occur with academic engagement is actually reflective of opportunities for enhanced learning.

In sum, I do think the authors have been generally responsive to prior feedback with important revisions made to the method and results section. There may be continued opportunity to strengthen the framing of this study in the introduction as well as enhance the alignment of the introduction with the discussion with regards to the major concepts addressed in this study.

Author Response

Please see attached word document that lists each reviewer comment and changes based on the reviewer recommendations.

Reviewer 4 Report

Comments and Suggestions for Authors

Dear Authors,

Thank you for your careful revisions and detailed response to reviewers. I appreciate the improvements you have made, and in this second round I provide feedback on how your revisions address the concerns raised in my initial review. I acknowledge the areas where the manuscript is now stronger, while also identifying points that remain insufficiently addressed or that still require clarification.

Abstract

  • First-round comment: I requested that the abstract avoid overstating the effectiveness of PMR and instead emphasize the exploratory nature of the study.

  • Revision: You have softened the language to “preliminary examination” and included a note on the small sample size and need for future research.

  • Evaluation: This revision is appropriate. However, some sentences (lines 24–25: “was an effective strategy to increased academic engagement… with moderate to large effect sizes”) still overstate certainty. Please maintain consistency by using tentative language throughout (e.g., “showed promising increases” rather than “was effective”).

  • First-round comment: I suggested acknowledging the limitations of social validity measures in the abstract.

  • Revision: You added a sentence noting that the data came from one teacher and brief student scales.

  • Evaluation: This addition is clear and appropriate.

Introduction

  • Positioning of the study: In the first review, I encouraged you to frame this as a preliminary examination rather than as an addition to the evidence base. You have revised the abstract and purpose statement accordingly. This is well addressed.

  • Mechanism of PMR and academic engagement: I noted that the rationale for PMR effects on engagement was not clear. You added more detail on potential mechanisms (lines 161–163, 283–295). This is an improvement, though the discussion could still benefit from integrating theory more clearly (e.g., whether PMR affects arousal, attention, or sensory regulation in ways directly tied to engagement).

  • Momentary Time Sampling references: I suggested that you contextualize MTS in the introduction. You chose to explain its rationale in the Methods section instead. This is acceptable, but the transition still feels abrupt. One bridging sentence in the introduction could help readers unfamiliar with MTS understand why it was later used.

  • Deficit vs. neurodiversity framing: In the first round, I asked you to acknowledge heterogeneity in self-regulation and avoid equating ASD with deficits or disruptive behavior. You have revised several passages (lines 39–62, 74–87) to highlight unmet needs and positive outcomes such as autonomy and peer relationships. This represents good progress. That said, some residual deficit-oriented phrasing remains (e.g., “deficits in self-regulation skills… resulting in minimal social inclusion” at line 11–12). I recommend ensuring that deficit language is consistently balanced with strengths-based perspectives.

  • Transition to PMR from yoga/mindfulness: You have now explicitly contrasted PMR with these alternatives (lines 107–151). This section is clearer, though it remains quite long; consider tightening slightly while emphasizing PMR’s feasibility advantages.

  • Critical appraisal of prior PMR studies: You added limitations (lines 275–295). This addition is appropriate.

  • Purpose statement and academic engagement as “universally positive”: I asked for more nuance. You responded that engagement is overall positive due to its correlation with fewer disruptive behaviors. While I understand your rationale, this still reads as somewhat dismissive of the concern. A brief acknowledgment that engagement is not inherently meaningful unless aligned with the student’s goals would resolve this tension without undermining your point.

Methods

  • Heading “Method” vs. “Methods”: Corrected appropriately.

  • Participants and Setting:

    • IRB approval is now described at the start — addressed.

    • You added information about classroom selection, operationalization of engagement, and the choice of ELA. These additions improve replicability.

    • However, as noted in my first review, demographic context is still limited. You have clarified that SES data were not collected, which is acceptable but should be explicitly noted as a limitation in the Methods, not only in the Discussion.

    • The fact that participants were teacher-selected is acknowledged in the Discussion, but it would be clearer to highlight potential selection bias also in the Participants section.

  • Inclusion criteria: You added criteria (lines 346–348), which addresses my concern.

  • Terminology: The acronym “ESE” is now written out — addressed.

  • Organization of Methods: You have reordered sections as requested. This improves readability.

  • Procedural detail: You expanded descriptions of teacher training, TA, and video use, including randomization and differences between videos. This is much improved.

  • Video engagement as gatekeeper: You explained that no sessions were excluded due to low engagement, and you added engagement data to the Results. This is acceptable.

  • Treatment integrity: You reported TI as 100% but indicated session-level data could not be retrieved. This remains a limitation; please acknowledge this explicitly in the Methods as well as in the Discussion.

  • Social validity: You expanded the teacher and student measures, though limitations remain (e.g., brevity of student responses). You have acknowledged this as a limitation. Adequately addressed.

Results

  • Magnitude of change: In the first round, I requested discussion of whether observed increases represent socially meaningful improvements. You now note that Tau-U data should be interpreted cautiously (lines 790–794). This is appropriate, but the point could be emphasized more strongly given that final engagement percentages remained relatively low.

  • Session counts and absences: You clarified why session numbers differ across participants (lines 638–639, 687–688, 719–721). This addition resolves my concern.

  • Graphs and video variation: You have included descriptions of video content and randomization. However, the graphs still do not clearly distinguish between different videos. Consider whether this information could be reflected visually or summarized more explicitly in figure notes.

Discussion

  • Scope of outcomes: You acknowledge that physiological/emotional outcomes were not measured and suggest this for future research (lines 890–892). Addressed.

  • Sample size and teacher selection bias: You note the limitation in the Discussion (lines 968–971). As mentioned above, this should also be flagged in the Methods.

  • IOA coverage: You have added this limitation (lines 1022–1026). Appropriate.

  • Social validity limitations: Acknowledged (lines 1027–1034). Appropriate.

  • Interpretation of engagement increases: You note the need for cautious interpretation. Addressed, though as suggested under Results, further emphasis on the modest absolute levels of engagement would strengthen the discussion.

Overall Evaluation

You have made substantial improvements in response to the first-round comments, particularly by clarifying procedures, adding methodological details, softening overstatements in the abstract, and incorporating a more balanced framing in the introduction.

Remaining concerns include:

  1. Some lingering deficit-oriented language that could be better balanced with neurodiversity perspectives.

  2. Inconsistent use of tentative language in the abstract.

  3. Need to explicitly note selection bias and missing demographic information in the Methods as well as in the Discussion.

  4. Treatment integrity reporting remains incomplete — this should be acknowledged more directly as a limitation.

  5. Graphical presentation could better reflect the variation in PMR videos.

I recommend further revision to address these points. Once these adjustments are made, the manuscript will be much stronger and suitable for publication consideration.

Author Response

Please see attached word document that lists each reviewer comment and the changes made to the manuscript based on reviewer recommendation. 

Round 3

Reviewer 1 Report

Comments and Suggestions for Authors

Thank you for the opportunity to once again review, “Using Progressive Muscle Relaxation to Increase Academic Engagement for Elementary School Students with Autism Spectrum Disorder”, submitted for consideration of publication in Behavioral Sciences. I was a reviewer on both of the previous submissions. Though I appreciate the authors’ continued efforts to strengthen this manuscript, each revision seems to present additional areas that, from my perspective, need attention prior to publication for a broad array of consumers.

Introduction

Each revision of the introduction seems to complicate the logic rationalizing this study and the contribution the authors believe it makes to the existing literature. As has been identified in prior reviews, the introduction still needs greater theoretical grounding, language that reflects an asset based characterization of students with ASD, tempered language about what is known that recognizes varied methodologies utilized in studies contributing to the current knowledge base, and clarity of purpose for this study relative to the work that was actually performed.

As an example, on page 3 line 118, the authors state, “Although yoga and mindfulness-based programs have been reported as socially valid…” yet do not provide a definition of what is meant by “socially valid”, which itself is another complex construct, make the statement as something that is certain despite later evidence in the paragraph describing otherwise, and make the statement without any supporting in-text citations. Then, the idea of social validity is discussed more on pages 5-6.  This contributes to the introduction becoming more disjointed and harder for the reader to fully follow the logic leading to the purpose of this study.

I’m going to encourage the authors to reconsider the entire introduction such that it should be written more like the introduction of a brief report. Efforts to concisely communicate in 2-3 pages what is most important for readers to understand about the contribution of this study to the existing literature might help to mitigate the persistent issues identified in the prior reviews. In addition, consistent with my comments associated with the discussion section, there needs to be better alignment between the introduction and discussion with respect to the major concepts – self-regulation, social validity, academic engagement, and challenging behavior.

Method

I continue to believe the authors have been responsive to prior feedback and requests for additional details.

Results

As identified in both of my prior reviews, there continues to be an issue with the description of the social validity data that has not yet been addressed. Please check the note for Table 4 in which it states that a rating of 3 represents a sad face. This is different from the in-text descriptions of the ratings. Please confirm what is correct and adjust accordingly. I believe the note for Table 4 needs to be changed, otherwise the interpretation of the data is incorrect.

Discussion

I’ll reiterate the same feedback I provided in my prior review that the authors did not address in the revision or respond to as part of their response with this revision. The authors continue to provide a discussion of their findings that do not connect back to self-regulation, which is emphasized in the abstract and introduction as part of the rationale for the study. The focus continues to stay exclusively on feasibility and academic engagement. With my continued reservations about the framing of the introduction, I encourage the authors to consider how self-regulation should be addressed more fully in the discussion section relative to the implications of their study and prior work, or, refocus the introduction just on active engagement by reflecting on observable behavior and internal processes that occur with academic engagement is actually reflective of opportunities for enhanced learning.

In sum, the method and results sections have improved through the revision process. From my perspective, there continue to be opportunities to strengthen the framing of this study in the introduction as well as enhance the alignment of the introduction with the discussion with regards to the major concepts addressed in this study.

Author Response

Introduction Comments

While we respect Reviewer 3’s opinions related to the introduction of the paper, we are not in agreement with the recommendations to reduce the introduction to 2-3 pages. We feel cutting down the introduction this dramatically would not align with the other two reviewer’s comments from the previous two rounds of reviews in which we made substantial revisions and additions to the introduction.

Currently the introduction flows to:

  1. Describe self-regulation and the behaviors associated with self-regulation.
  2. Proceeds to discuss strategies that may improve self-regulation that are more common in the literature (such as self-monitoring).
  3. Then describes relaxation strategies like yoga and PMR that also have been used more recently to improve self-regulation.
  4. Discusses the research related to PMR in more depth to provide a rational for the use of PMR in the classroom including some advantages related to PMR
  5. Then describes some limitations in the research specific to the lack of direct observation data on classroom behaviors (academic engagement) using PMR.
  6. This leads to the purpose of the current study to conduct a preliminary investigation using direct observations of academic engagement using PMR in the classroom for students with ASD.

We feel the flow of the introduction leads well to the purpose of the study and are hesitant to drastically reduce the introduction as this would not align with prior comments from other reviewers. We did modify the sentence on page 3 to remove the phrase socially valid as we agree that we did not discuss this construct yet in the paper.

Results Comments

We changed the note in Table 4 to include the correct representations for the social validity ratings. Thank you for catching this.

Discussion Comments

We added some information early on in the discussion section to hopefully better tie in self-regulation as we agree this should be mentioned in this section. Perhaps some of the confusion is that we see improved academic engagement as an improvement in self-regulation skills. In the introduction, on p 2 (lines 70-79) we describe that research indicates that the benefits of improved self-regulation include improved engagement. Therefore, when we are discussing the improvements in academic engagement this aligns with improved self-regulation. We did make some additions on Page 16 lines 942-946 and Page17 lines 1041-1046 to specifically state self-regulation and its relation to academic outcomes. We hope this helps with any confusion or issues with the alignment of the discussion section to self-regulation.